# Research on the Influence of Different Designs on Mindfulness Meditation in Virtual Reality

Xinyu Cui*

Tsinghua University

## ABSTRACT

Mindfulness meditation in virtual reality can play a role in regulating emotions in daily life, but there are few studies on the different designs and interactions of mindfulness meditation in virtual reality(VR). This study focuses on the design and influence of different mindfulness meditation systems based on VR. From three aspects of virtual environment design, virtual object design and interactive task design, we designed specific schemes of mindfulness meditation systems in VR and completed the development of different systems. We evaluated different designs qualitatively and quantitatively through four sets of comparison experiments. Pulse signals and electroencephalography(EEG) signals were collected for physiological measurement, while the Brunel Mood Scale ( BRUMS) and questionnaire were used for subjective assessment. In the analysis of the results, we obtained the design guidance of the mindfulness meditation system in VR and discussed the possibility of other designs.

**Index Terms:** Human-centered computing—Human computer interaction (HCI)—Interaction paradigms—Virtual reality; Human-centered computing—Human computer interaction (HCI)—Empirical studies in HCI

## 1 INTRODUCTION

In psychotherapy, the concept of mindfulness is applied to help people deal with their disorder or cope with life, often guided by a therapist or technology. The popularity of mindfulness meditation has continued to rise since the COVID-19 pandemic, and it is gaining more attention as an effective way to heal the body and mind.

The concept of mindfulness first appeared in Buddhism and was introduced into psychological research in the 1970s. Jon Kabat-Zinn introduced mindfulness into therapeutic practice, and since then hundreds of medical centers, hospitals and clinics around the world have adopted it to reduce pain and stress.The use of mindfulness in clinical Settings has proven to be an effective tool to help patients cope with disorders such as depression, anxiety, and obsessive-compulsive disorder. At the same time, as mindfulness is a mental training method with very low space and time limit, people can also practice mindfulness in their daily life to relieve pressure and negative emotions.

Meanwhile, with the rapid development of VR technology and hardware, VR can bring users a better sense of immersion. Theoretically, there is a strong link between flow in mindfulness and immersion in VR, so some psychology researchers have applied VR to mindfulness meditation. Research has shown that VR systems and apps can help guide mindfulness or improve people's mindfulness skills, thereby improving well-being. In addition, mindfulness in VR has a good therapeutic effect on many patients with mental illness.

In recent years, the number of human-computer interaction (HCI) studies around mindful meditation has shown a steady increase.

---

*e-mail: cui-xy20@mails.tsinghua.edu.cn

For mindfulness design in HCI, a few people have proposed some mobile digital mindfulness design frameworks and models, but few people have studied the impact of different designs on mindfulness meditation in VR.

So far, we can put forward the following thoughts based on previous studies: **How can we better design a VR world of mindful meditation? And how do we resolve the tension between the stillness required for mindfulness and the motion required for interaction?**

Therefore, according to the BehaveFIT framework proposed by Wienrich et al. [44], we studied the design of mindfulness meditation in VR from three aspects: virtual environment design, virtual object design and interactive task design.

**Virtual Environment Design:** Virtual environment, as the main carrier in VR, provides a wide range of possibilities for creating different emotional frameworks for mindful experience. Instead of having to construct a scene through your own imagination, you can use a VR device to put yourself in a new scene. As VR devices continue to upgrade, there are a lot of well-made games that are extremely realistic and immersive. At the same time, some VR games use a cartoonish flat modeling style to put the player in a different dimension of the world. Therefore, the following aspects of virtual environment are selected for design research: (1) flat style, (2) realistic style.

**Virtual Object Design:** Self-regulation of attention is a very important part of mindfulness. The object of attention can be tangible or intangible objects such as experiences, objects and thoughts. It's much easier to focus on the tangible objects in the scene in VR. In the VR meditation system RelaWorld [18], researchers use a graphical interface and floating objects to guide users to focus. YU, developed by Zhu et al. [49], also designed a koi fish pond that simulates nature to experience and receive the changing states of participants' bodies. The fish moves as the only living thing in the pond, allowing users to focus their attention. In both VR meditation systems, the researchers designed different virtual objects to direct attention. Therefore, virtual objects will be designed from the following aspects: (1) static virtual objects, (2) dynamic virtual objects.

**Interactive Task Design:** Most traditional mindfulness or meditation tasks involve becoming aware of and experiencing your body, such as sitting, body scanning meditation, and mindful walking. These exercises create body awareness that is closely related to mindfulness. Most of the tasks in existing digital mindfulness systems involve speaking in a calm environment to direct the user's attention to visual cues, breathing, or physical sensations. Only a few studies have shown more active dynamic interactions in virtual environments, such as the mobile app PAUSE [26], which asks users to move a finger slowly, continuously, and repeatedly across the screen. In VR, even the simplest virtual environments offer some kind of interactivity, a variety of possibilities and freedom in interaction design. PsychicVR [2]allows users to create or light flames in a 3D environment while they are focused. We will explore the following two aspects of interactive tasks: (1) static body scanning task and (2) dynamic body motion task.

To investigate the above issues, we designed four mindfulness meditation systems in VR. In the development process of this study,

Unity 3D software was used to build the VR environment, and PICO Neo3 Pro VR equipment was used to allow participants to interact with VR.

We recruited 15 participants to conduct mindfulness meditation experiments under different experimental conditions. During the process, BrainLink EEG headband and pulse sensor were used to measure and calculate physiological indicators to evaluate the level of meditation relaxation and attention in different experimental groups. Subjective questionnaires were used to investigate the participants' preference and immersion in different experimental conditions. BRUMS were also used to measure mood changes before and after the experiment to assess the overall effect of the system.

## 2 RELATED WORK

### 2.1 Mindfulness Meditation Research

In the late 1970s, Kabat-zinn separated the practice of meditation from Buddhism. He defines mindfulness as follows: "Mindfulness is a consciousness that emerges through consciously, in the present moment, without judgment, focusing on the unfolding of each moment of experience" [17]. Since then Mindfulness-Based Stress Reduction Therapy (MBSR), Mindfulness-Based Cognitive Therapy (MBCT), Dialectical Behavior Therapy (DBT) and other therapies have been used to treat such as chronic pain [22], depression relapse [36] or borderline personality disorder [21].

Baminiwatta et al. used bibliometric analysis to analyze the articles related to mindfulness retrieved from the Web of Science. [3] According to the analysis, in the 55 years since the first publication on mindfulness was published in 1966, 16,581 articles have been written about mindfulness. Since 2006, publications on mindfulness have grown exponentially. 47% of publications are psychological and 20.8% are psychiatric. From 1966 to 2015, most of the earliest studies were on outpatient programs, reflective exercises, and transcendental meditation. During this period, the most researched directions were mindfulness therapy intervention and meditation practice. Several different directions have also developed from mindfulness therapy interventions, such as MBSR, MBCT, acceptance and commitment therapy, mindful yoga, and self-compassionate therapy. From 2016 to 2021, the main research direction is modulation, reflecting a recent increase in interest in the mechanisms and moderating variables behind mindfulness interventions. At the same time, the application of mindfulness in the management of a variety of clinical conditions has become increasingly prominent, such as eating disorders (mindful eating), addiction (smoking cessation), and bipolar disorder. And with the outbreak of COVID-19, the latest research on mindfulness apps, online mindfulness interventions, reducing loneliness and more is growing.

### 2.2 Virtual Reality Therapy

In the medical field, many disciplines try to apply VR system to diagnosis, treatment and other fields. Especially in psychiatry, where traditional treatments are limited to interpersonal psychotherapy and medication, VR can provide various types of stimulation [24]. Stress and pain can have detrimental effects on the body and mind, and some experimental results suggest that VR technology has practical benefits for subjective pain reduction [14]. So VR can improve the mental disease in patients with post-traumatic stress disorder [35, 45], phobias, anxiety [25], depression [11], cognitive and social functioning [23].

### 2.3 Mindfulness in Human-Computer Interaction

In the definition of mindfulness in HCI, the most mentioned aspects are attention, presence, experience (including experience, thoughts, feelings, etc.), non-judgmental, immediate and conscious. There has been a detailed review of HCI and mindfulness. They have analyzed and clustered 38 articles to establish a classification framework for HCI research in the context of mindfulness. The framework includes four different mindfulness perspectives in HCI: role, practice style, duration, common aspects, and seven common research topics: meditation practice, therapy, Reflection and knowledge Acquisition, Mindfulness in everyday life, Mindfulness in interaction, task performance enhancement, and meta-level research [38]. Many researchers have developed and applied various types of mindfulness technologies (such as entity [4, 39, 43], mobile app [6, 40], and VR system [12, 18, 27, 32]) to help users gain and maintain attention to internal or external experiences for facilitating, enhancing, or simulating meditation practices.

Currently, a number of models and frameworks exist that define guidelines and frameworks for digital mindfulness design in smartphone apps and wearables. Niksirat et al. proposed and validated an interactive Mindfulness-Based Mobile Application (MBMA) framework that provides the basis for the design of interactive meditation. [30] Zhu et al. proposed a model that addresses and aggregates types of digital mindfulness support, consisting of four successive stages. They came up with specific design principles: focus only on the present, engage with everyday life, and don't evaluate accepted aesthetics. [49] In addition to the study of the overall design framework, some researchers have discussed design elements for specific scenes, such as the design of background sound [8].

In recent years, many other researchers have also worked on virtual mindfulness design [10]. Such research mainly uses technologies that combine the virtual and the real world, including virtual reality (VR), augmented reality (AR), mixed reality (MR), collectively known as extended reality (XR). The BehaveFIT framework proposed by Wienrich et al describes the direct and indirect effects of virtual technological interactions on human perception and behavior [44]. They also define virtual reality in terms of virtual environments, virtual objects, virtual others, and virtual self-representations. Roo et al. developed a set of guidelines to support design based on XR mindfulness [29]. The two guidelines they put forward that most specifically apply to XR design are: use tangible interactions, and choose the right reality. In order to ensure that users pay attention to their bodies, they promote tangible interaction and tactile feedback, in addition to emphasizing the importance of considering users' personal characteristics regarding XR perception.

## 3 MINDFULNESS SYSTEM DESIGN

We wanted to explore how to better design a mindfulness meditation system in VR, focusing on four sub-questions:

**Q1:** How do video and VR affect mindfulness meditation?

**Q2:** How do different virtual environment designs affect mindfulness meditation in VR?

**Q3:** How do different virtual object designs affect mindfulness meditation in VR?

**Q3:** How do different interactive task designs affect mindfulness meditation in VR?

If we want to study the above problems, we need to construct a virtual system suitable for mindfulness.

In terms of hardware, VR headsets have the advantage of blocking out external distractions, while visual cues in VR can direct the user's attention in a more subtle way than just sound guidance or small screen visual guidance. However, any visual reference can distract users from focusing on themselves. As a result, VR needs to be carefully designed to ensure attention and not create new distractions through overly complex designs. At the same time, VR is far superior to other hardware in terms of immersion. The term immersion defines the extent to which a computer display is able to convey to the senses of human participants the illusion of inclusive, extensive, ambient and vivid reality. Especially considering that the concept of presence is a major feature of VR's specific perception, and presence is a recurring feature of mindfulness. In order to create immersion in a virtual environment, the media should isolate the

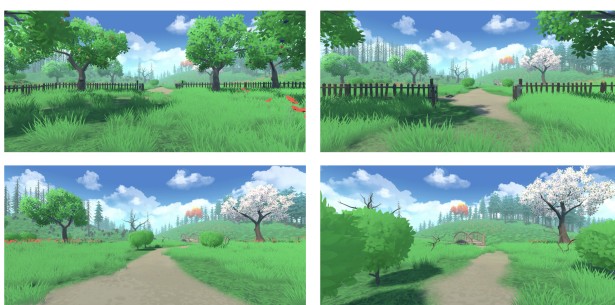

Figure 1: Flat Natural Virtual Environment In VR, the first person perspective changes as the camera moves slowly in the scene. This scenario is also used in video mindfulness meditation.

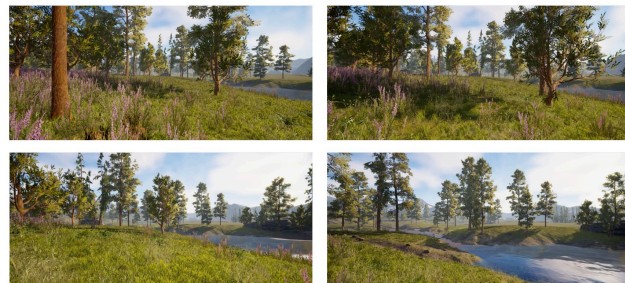

Figure 2: Realistic natural virtual environment In VR, the first person perspective changes in the picture.

user from their physical environment, and the user's actions should have consequences in the virtual environment.

In the design of the mindfulness meditation system in this study, we combined the immersion of virtual reality, the characteristics of mindfulness, and the research question.

### 3.1 Scene Design

Mindfulness is derived from Buddhism, and it is at the core of all classical Buddhist meditation systems. In the existing VR meditation, many researchers have designed VR environments based on Buddhist elements. For example, multiple users perform traditional compassion meditation [15] in a VR environment at the same time. The setting consists of six stone statues that sit on a small platform resembling a temple. Meanwhile, according to the experience of some mindfulness meditation researchers [42], meditative awareness can be used to enhance the restorative and balancing qualities of nature, while spending time in nature can in turn enhance meditative awareness. Other studies have shown that natural environments can enhance mindfulness programs through comparative experiments [7].

In this study, the theme of the scene of mindfulness system was determined as nature and Buddhism. Q1 and Q2 will use natural scenes, and Q3 and Q4 will use Buddhist scenes. Because nature scenes will be used to study the different effects of flat style and realistic style scenes in virtual environment in VR. So we need to build two different styles of VR nature scenes – flat nature scenes and realistic nature scenes. Since the Buddhist scene will be used to study the different effects of different virtual objects and interactive tasks in the VR mindfulness system, from the perspective of experiment, we only need to construct a Buddhist virtual scene, and then change the specific design details in the scene according to different research directions.

In flat natural scenes, mainly choose the different types of trees as a scene in the main scene elements, also used the stones and timber, Bridges, streams and other elements. In the selection of the model, the low precision model is mainly used. This kind of model is characterized by simple structure, fewer facets and fewer details. Also, their materials are mostly solid colored material balls, with no extra texture maps. The whole scene is divided into two parts by the stream, connected by a small bridge over the stream. On one side was a wood-fenced garden, which led to a path that led to a stream with cherry trees. On the other side is the hillside, which is covered with a lot of vegetation.

In the realistic natural scene, the model mainly uses high-precision model, which has complex structure, fewer facets and rich details. In order to get closer to the physical environment, lighting and texture mapping were needed to simulate the trees in the real world. The overall layout of the scene is mainly composed of a forest of trees with different details, and a clear blue lake. In the distance, a snowy mountain is placed as a distant view to increase the sense of hierarchy of the scene.

Finally, in the Buddhist scene, we used a large stone statue to become the visual center of the whole, with a long candle in its hand. At the same time, some models such as stone columns and dragon-shaped stone sculptures are also placed in the scene to enrich the details of the Buddhist scene. Models of cherry trees and maple trees were also used to enhance the overall atmosphere. Ponds, rocks and other elements are added to enrich the details of the scene. The scene also incorporates parts of Japanese Zen architecture. In the layout of the Buddhist scene as a whole, a platform centered on the Buddha statue is constructed, and different platforms are connected by stone steps. The basic colors of the models were mainly red and yellow, which are commonly seen in Buddhist temples. At the same time, the tone is set to warm color, which is more in line with the solemn and bright feeling of Buddhism. In this study, different virtual object designs are studied in Buddhist scenes. After preliminary testing, it was decided that the static virtual object in the scene would be a stone statue, while the dynamic virtual object would be an elderly monk holding a wooden pestle and slowly tapping a wooden fish. The monk, dressed in a red robe, sits in front of the statue and blends in nicely with the scene. Compared with other more abstract objects such as flames, the stone statue is more consistent with the shape of the monk. Moreover, the movement of tapping wood fish is small and regular, which can better meet the external environment required by mindfulness meditation.

### 3.2 Interaction Design

Most of the current VR mindfulness designs are static interactions that don't require any physical movement by the meditator. For the static body scanning task, this study set it as the simplest breathing exercise.

Designing a dynamic interaction that works well for VR is more challenging. When thinking about what kind of dynamic interaction methods and techniques can be applied to VR mindfulness practice, we refer to the relaxation response principle [13], which refers to the relaxation response as a physical state of deep rest. According to the relaxation response, slowly repeating an action helps the practitioner release chemicals and brain signals that relax the body and stabilize the mood. Relaxation response slow speed requires practitioners to focusing on the present through ignore daily thoughts. A relaxation response can be triggered by slowly repeating words, sounds, breaths, or actions. Body movement produces interoception, kinesthesia, and proprioception. It is worth noting that moving the body at a slow speed enhances these sensations and requires the user to pay attention to the body movement in the moment [31].This reflects the common feature of Tai Chi, yoga, Qigong and walking meditation, which is based on slow, continuous and gentle movements.

In Eastern forms of meditation, there are many use cases utilizing embodied cognition, such as the Buddhist rosary, the Tibetan prayer

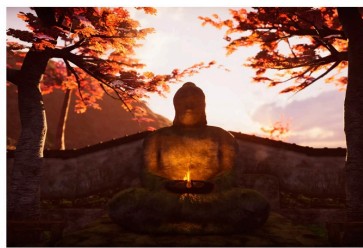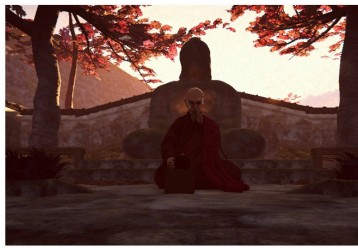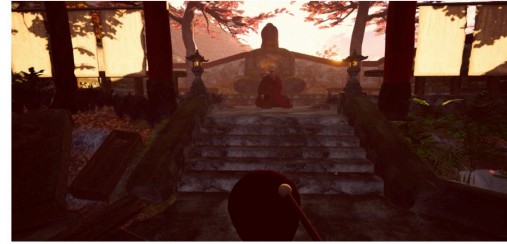

Figure 3: The left picture is a static virtual object, which is a stone statue. The middle image is a dynamic virtual object, which is a monk slowly tapping a wooden fish.The right picture for the use of VR handle to interact with a first-person perspective.

wheel, the Chinese Meditation ball, and the Tibetan Psalm Bowl, all of which use simple tangible artifacts to direct and regulate attention through body movements. We noticed that the mallets used to hit the wooden fish have a similar shape to the VR gamepads, and as a result, they have a similar grip.The hand action of tapping also well meets the slow repetitive action suggested in the relaxation response principle mentioned above. Therefore, this study sets the dynamic body movement task as simulating tapping on the wooden fish through the handle in VR. The first-person tapping on the wooden fish in VR is shown in Fig. 3.

### 3.3 Sound and Animation

In terms of sound design in this study, it can be divided into two parts: guided audio in mindfulness meditation and background ambient sound in virtual reality.

In mindfulness meditation, voice guidance is the most common form of guidance. The key to guidance is to use words to quickly focus the meditator's attention on what we want him to pay attention to. You can also act as a guide or coach to teach step-by-step meditation techniques. According to the required mindfulness tasks in the system, in order to meet the requirements of subsequent experiments, we need three guiding audio: (1) breathing exercises; (2) Breathing exercises followed by body scans;(3) Guide the tapping of wooden fish and breathing exercises. In this study, a large number of mainstream meditation guidance words were collected, and the final text of the guidance words was obtained after screening and adjustment. Then I found a professional announcer for dubbing and recording to get the final guide audio, and the length of each guide audio was controlled within 4-6 minutes.

In VR, background sounds in the environment are also a key factor in increasing immersion. We used binaural beat audio. Binaural beat refers to the phenomenon that occurs when two sinusoidal waves of different frequencies are transmitted into both ears at the same time. The brain adjusts its frequency to match that of the binaural beat. Binaural beats can significantly alter the frequency of our brains to achieve the desired state. This state can be a calm, relaxed state, or a state of wakefulness and alertness. The slightly different frequencies generated by the two sources in the binaural beat are called beat frequencies, and the maximum frequency that can be used to generate beat frequencies is about 1000-1500 Hz, and the maximum beat frequency that humans can respond to is 30 Hz. One study confirmed that different beats produce different brain responses. [16] It has also been shown that when binaural beat audio is used during meditation practice, the power spectrum of Theta (5 Hz) and lower Alpha (8 Hz) increases and propagates in the midfrontal line of the brain. [46]

Therefore, in this study, 5 Hz beat frequency was used to activate the Theta band and 150 Hz carrier frequency. And chose melodies with little fluctuation in pitch as background music. Two binaural beat background audio files were eventually generated for use in subsequent experiments. In addition, some forest background sounds

are also used in the natural scenes, such as bird chirping, cicada chirping, wind sound, and water flow sound. Buddhist scenes also feature bells and the sound of wooden fish.

In the animation side, it mainly increases the animation effect for the objects in the virtual environment, so as to achieve the purpose of more realistic. Such as the slight shaking of trees and the fluctuation of water in the scene. Flickering candles and animations of monks were also added to Buddhist scenes.

## 4 EXPERIMENT

The study carried out five different experiments in VR and one experiment in video.

**Experimental design** According to the problems of this experimental study, it can be divided into experimental group A and experimental group B according to the scene and control relationship, and there are three specific experiments in each group. The specific content is shown in the figure. Firstly, we recruited 15 people (5 females) from Tsinghua University to participate (age: M = 22.933, SD = 1.534, range = 21-26).Seven participants reported having meditated in the past. Each participant was compensated $10.

Before the experiment began, each participant signed an informed consent form, which specified the purpose of the experiment, the procedure, and the associated risks. Participants were also introduced to how the VR program was used and allowed to try out the VR device. At the same time, pre-test questionnaires were filled in before the experiment. Then, participants wore physiological sensors and conducted mindfulness meditation experiments according to the sequence randomly generated in the program. Participants' EEG signals and pulse signals were measured and recorded to assess levels of relaxation and concentration. Each scenario was meditated for about 4-6 minutes, with a 5-minute break between each scenario, and the entire experiment lasted about an hour. Some scenarios required participants to interact with VR controllers. At the end of all experiments, the subjects were also required to fill in the post-test questionnaire.The experimental process design is shown in Fig. 4.

The site of the experiment was the workshop of the teaching building of Tsinghua University, and the environment during the experiment was shown in Fig. 5. Participants sat in a regular armchair and measured their pulse with finger clips, measured EEG with a headband, listened to audio through a wired headset and wore a VR headband during VR mindfulness meditation. When interacting with the handle, participants were allowed to rest their elbows on the arm of the chair to prevent fatigue when their hands shook the handle.

**Questionnaire** The subjective questionnaire used in this study includes pre-test questionnaire and post-test questionnaire.

The POMS version of question 65 requires relatively long filling time [9], which brings great burden to respondents. To address this issue, the researchers developed several simple versions of the BRUMS with 24 questions [37], which retained the original POMS

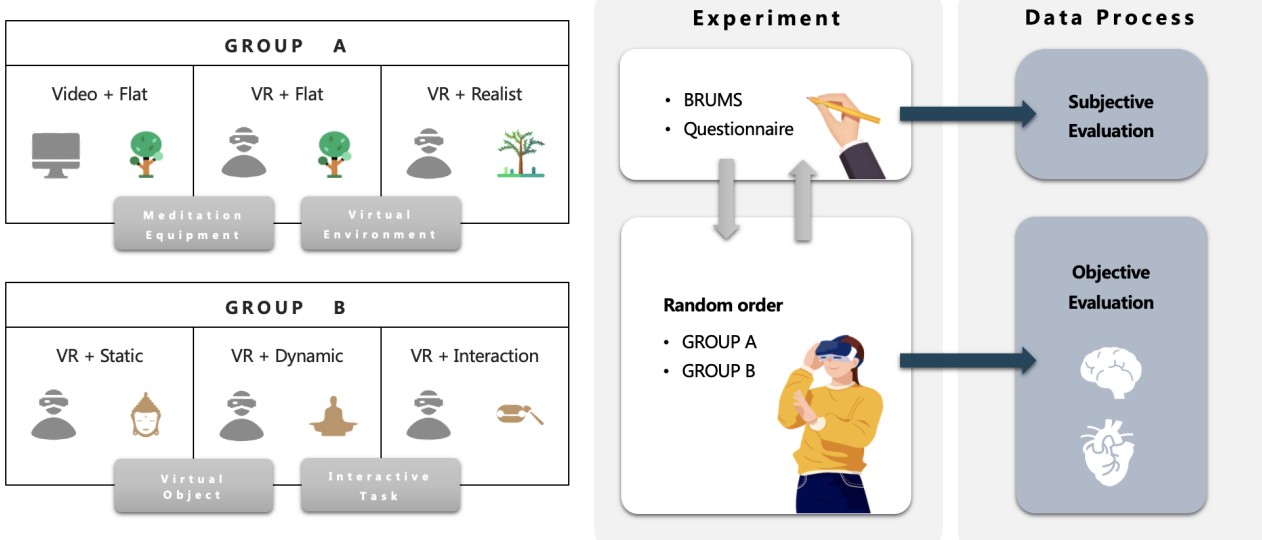

Figure 4: The Settings of different experimental groups and corresponding research questions are shown on the left.The flow in the user experiment is shown on the right.

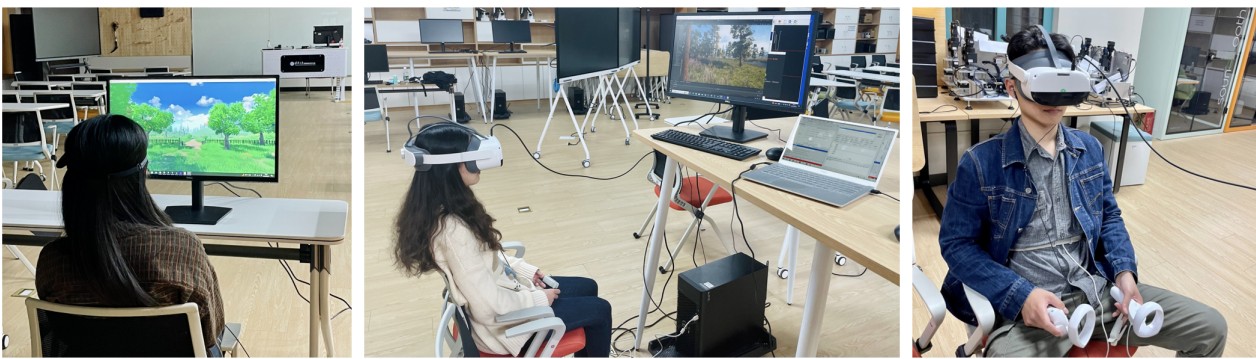

Figure 5: Participants in the figure performed video mindfulness meditation, realistic virtual environment meditation and dynamic interaction respectively.

measurement structure and included six subscales: anger, confusion, depression, fatigue, tension, and vitality. Because of its reduced completion time and powerful psychometric properties, BRUMS has been widely used in the field of sports and exercise and for a variety of purposes.

Uses of BRUMS in the field of exercise include assessing emotional responses to various types and intensities of exercise, and monitoring emotional responses to music. In addition to the sports and exercise field, BRUMS are used to screen for post-traumatic stress disorder [41], monitor the mental health of cardiac rehabilitation patients [19], and assess adolescents at elevated risk of suicide [1]. Therefore, there is a lot of evidence supporting the utility of BRUMS in application and clinical Settings. In addition, BRUMS is one of the few mood measures that has been translated into multiple languages and validated across cultures, the scale has been validated in languages such as Chinese [48], Italian [28], Malay [20], Spanish [5], etc.

The order of description words in BRUMS is scrambled. The subjects need to choose the scale that is in line with their own experience. There are 5 scales (0-4). The sum of the scores of each sub-scale gives a sub-scale score ranging from 0 to 16, and the

questionnaire will not produce an overall score. Subjects fill in the BRUMS each time generally only 1-2 minutes to complete.

Among them, the pre-test questionnaire includes the basic information of the subjects and BRUMS. Basic information included the subject's name, gender, and whether or not they had meditated, and the time limit for the BRUMS measure was "last week". The post-test questionnaire included BRUMS and subjective experience questions. The time limit of BRUMS measurement is "now". Subjective experience questions include ranking questions and open-ended questions, which are used to investigate the user experience of the subjects. The ranking questions asked participants to rank six different meditation experiments in terms of liking and immersion, while the open questions asked their subjective evaluations of the VR mindfulness meditation system.

Physiological measurement    According to previous findings, mindfulness exercises can affect the autonomic nervous system of users, thereby unconsciously regulating bodily functions. We can monitor and evaluate the performance of the VR mindfulness system by measuring physiological indicators [34, 47]. Based on the characteristics of VR and considering that the device needs to be worn for a long time, pulse measurement and EEG signal were

chosen in this experiment.

Heart rate is calculated in beats per minute (bpm). Because the pulse of a normal person is consistent with the heart rate, we can calculate the heart rate by measuring the pulse. Pulse can be measured using infrared light. Infrared pulse sensor transmits infrared light with a specific wavelength to the finger tip, collects the signal of blood volume change in the last little blood vessels, and outputs the complete digital signal of volume wave. The signal has strong similarity to the piezoelectric pulse signal, which can be used for emotion analysis. The pulse sensor used in this study uses a USB communication interface with a baud rate of 57600Hz and a sampling frequency of 200Hz. When in use, the finger clip of the sensor is clamped on the index finger of the subject, and then the lead wire is connected to the host box and connected to the computer. Install the USB driver of the sensor and use the serial port for communication. Select the serial port to receive data from the infrared pulse sensor. The data transmitted includes the start bit, data bit, and stop bit. Pulse amplitude data of each sample was transmitted with 4 hexadecimal digits. In the experiment, we saved these data locally for later data analysis.

EEG signals measure the electrical activity generated by the synchronized activity of thousands of neurons. Because the voltage fluctuations measured on the electrodes are so small, the recorded data is digitized and sent to an amplifier, which can then display the amplified data as a series of voltage values. EEG signals are usually studied in five major bands: Delta (0.5 - 4Hz), Theta (4 - 7Hz), Alpha (8 - 13Hz), Beta (13 -30Hz), and Gamma (30 - 45Hz).

In this study, a non-invasive technique was used to record electrical activity generated by the brain using electrodes placed on the scalp surface. EEG signals were recorded using BrainLink Pro EEG consumer-grade headbands. According to the International 10-20 system, the BrainLink band consists of two active electrodes at FP1 and FP2 in the frontal lobe. The BrainLink Pro EEG headband samples ECG signals at 512 Hz, filters the signals, and then uses a peak detection algorithm to measure and check artifacts. After the acquisition and processing of the original data, the spectrum analysis of the signal is carried out, which needs to calculate the power spectrum of the signal. The power spectrum can be calculated directly from time domain signals.

$$P_f = \frac{1}{N} \left| \sum_{n=0}^{N-1} x[n] e^{-j2\pi fn} \right|^2 \qquad (1)$$

After calculating the power spectrum of the EEG signal, the waveforms are subdivided into Delta, Theta, lowAlpha, highAlpha, lowBeta, highBeta, lowGamma, middleGamma. BrainLink Pro transmits data over Bluetooth 4.0 at 1Hz. Using a BrianlinkSDK on a computer to receive EEG data, After the connection is complete, EEG data is captured and stored for subsequent analysis.

## 5 DATA PROCESSING AND DISCUSSION

### 5.1 Data processing

When the pulse wave passes under the sensor, the signal value rises rapidly, and then the signal falls back to normal. Since the PPG signal is repetitive and predictable, we can select any recognizable feature as a reference point, such as calculation by peak detection. However, because the amplitude of PPG signal will change and fluctuate, there will also be a lot of noise in the signal. So, we can filter the signal first. Butterworth filter is a commonly used low-pass filter, that is, to allow the low-frequency portion of the signal to pass through. It is characterized by a very uniform response to the frequency within a specified range, and is the filter with the flthest amplitude. A fifth-order Butterworth filter with a cutoff frequency of 2.5Hz is used in this study. After filtering, we can calculate the dynamic threshold curve of PPG signal by moving average, and then use the peak value to determine the period of PPG signal. Finally,

we can calculate the value of heart rate. After the experiment, the stored pulse signals were used to calculate heart rate, and the next four minutes of data were taken, The average heart rate is calculated according to the above algorithm by sliding the window for another 10 seconds. The window slides at the rate of 2 seconds, and the obtained array is processed with outliers. So we can get an array of changes in the center rate of the experiment.

Because we did not find an effect on average heart rate, the heart rate range was used for the analysis. The heart rate range is the overall distribution of heart rate during the experiment, which is shown by boxplot. The increase of heart rate range corresponds to a better state of relaxation. Lorem ipsum dolor sit amet, consetetur sadipscing elitr, sed diam nonumy eirmod tempor invidunt ut labore et dolore magna aliquyam erat, sed diam voluptua. At vero eos et accusam et justo duo dolores et ea rebum. Stet clita kasd gubergren, no sea takimata sanctus est Lorem ipsum dolor sit amet. Lorem ipsum dolor sit amet, consetetur sadipscing elitr, sed diam nonumy eirmod tempor invidunt ut labore et dolore magna aliquyam erat, sed diam

In this study, brain waves in the Theta and Alpha bands were selected. In the bands of EEG signals, increased activity in the Theta band was associated with meditation concentration, while increased activity in the Alpha band indicated relaxation. Previous work has also looked at activity in low Alpha (8-10Hz) and high Alpha (11-13Hz) bands [33]. They analyzed changes in psychophysiological parameters in 20 normal adults during meditation, and the results showed increases in Theta and low Alpha bands during meditation.

For EEG data in each experiment, the corresponding proportion of Theta, Low Alpha and High Alpha frequencies in the total power (0-45Hz) was calculated by using the absolute value of the power, ranging from 0-100%.The resulting timing signals were then moving-averaged to eliminate the effects of high frequency noises such as movements and blinks.Since we wanted to observe changes in the power ratio of different frequency bands before and after meditation, we evaluated changes in EEG signals reflecting changes in mindfulness meditation states by taking the difference between the power of different frequency bands at the beginning and the end of meditation. Each experiment intercepted the same point in time.

The ranking questions in the questionnaire asked participants to rank six different meditation scenarios in descending order of liking and immersion. Liking is used to evaluate the user experience, and immersion is used to evaluate the effect of using VR. We set different scores for different rankings, ranking from low to high will be 1-6, so we will get the score data of 15 participants. We evaluated the score of each experiment to obtain subjective evaluation of the liking and immersion of different experimental scenes.

SPSS software was used for data analysis in this study. For the hypothesis test, all EEG data passed the homogeneity of variance test and the normality test. We compared the effects before and after mindfulness meditation in different experimental groups using repeated measure Analysis of variance (ANOVA), with significance set at $p = 0.05$. However, the pulse rate data did not pass the parameter evaluation test. Therefore, Mann-Whitney U Test was used for nonparametric analysis.

For the BRUMS scale rating data, we sum the scores of each subscale to produce a subscale score ranging from 0 to 16. In this way, we can get six dimensions of mood change for each participant before and after the experiment as a whole. Then, the overall evaluation of the effect of VR mindfulness meditation was obtained by calculating its mean value.

### 5.2 Results and Discussion

Video and VR Mindfulness Meditation    The experimental results are shown in Fig. 6 and Table 1. In terms of EEG signals, the power increase of signals in Theta, Low Alpha and High Alpha bands of Video group and VR group is lower than that of VR group.

Table 1: Variance analysis in different equipments

| | Video | VR | F | p |
|---|---|---|---|---|
| Theta | 0.26±2.61 | 1.10±3.54 | 1.086 | 0.302 |
| Low Alpha | 0.82±2.00 | 1.59±1.54 | 2.816 | 0.099 |
| High Alpha | 1.40±1.57 | 2.50±1.40 | 8.251 | 0.006** |
| Heart Rate | 82.44±12.64 | 81.23±13.04 | 836776 | 0.000** |
| Liking Degree | 2.13±1.88 | 3.33±1.45 | 3.825 | 0.061 |
| Immersion | 1.33±1.05 | 2.93±1.28 | 14.049 | 0.001** |

\* p<0.05 , ** p<0.01

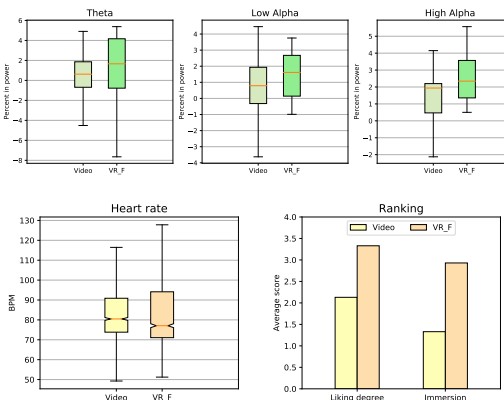

Figure 6: Physiological indicators and subjective scores in different equipments

Among them, the results of the video group and VR group show no significant difference for Theta and Low Alpha band, while the results of different groups show significant difference for High Alpha band (p=0.006). To be specific, The video group's average increase in High Alpha band (1.40) was significantly lower than that in VR group (2.50). In terms of pulse signals, there was a significant difference in heart rate distribution between the video group and the VR group in the mindfulness meditation experiment (p=0.000), and the VR group had a lower overall heart rate distribution. In terms of the upper and lower quartiles, the VR group had a wider range of heart rate, indicating that the heart rate decreased more before and after mindfulness meditation. In terms of subjective rating, the VR group scored higher on liking and immersion(p=0.001) than the video group.

In conclusion, from the perspective of physiological signals and subjective experience, we can conclude that VR mindfulness meditation has better effects than the video group.The results are consistent with previous research conclusion, VR in meditation can better help people to relax. At the same time, participants also reported that VR modeling scenes have a strong atmosphere and can bring a good sense of immersion. This finding also encourages more developers to use VR to design and develop apps for mindfulness meditation or relaxation.

Flat and Realistic Virtual Environments   The experimental results are shown in Fig. 7 and Table 2. In terms of EEG signals, the power increase of signals in Theta, Low Alpha and High Alpha bands of the flat group is higher than that of the realistic group, but there is no significant difference. In terms of pulse signals, the overall heart rate distribution was slightly lower in the flat group (p=0.026), but there was no significant difference in the heart rate range between the two groups in terms of the upper and lower quartiles. On the subjective scale, there was no significant difference in likeability scores between the two groups, but the realistic group scored higher on immersion than the flat group (p=0.003).

In summary, we can think that the flat style virtual environment is not very different from the realistic style virtual environment. In terms of physiological signals, the flat group was slightly better than the realistic group. From the perspective of subjective experience, the realistic group is slightly better than the flat group. We can infer that for mindfulness meditation, a more concise style can play a good role, which is very beneficial for VR devices. Developers can reduce the accuracy of the model to ensure proper operation in VR devices. There was also a lot of subjective interest in the realistic virtual environment, which we hypothesized might be due to the fact that the rich details in the virtual environment attracted some of the participants' attention, which resulted in mediocre performance on the EEG signals. However, participants in this study generally said that the picture quality of VR was not good enough to represent the details of realistic scenes, so the results were limited by the resolution of the VR devices used. We still believe that a more realistic design would be easier to leverage VR's strengths, but it would require more careful mission and scene design.

Table 2: Variance analysis in different virtual environments

| | Flat | Realistic | F | p |
|---|---|---|---|---|
| Theta | 1.10±3.54 | 0.66±2.31 | 0.322 | 0.573 |
| Low Alpha | 1.59±1.54 | 1.42±1.30 | 0.218 | 0.643 |
| High Alpha | 2.50±1.40 | 2.12±1.95 | 0.765 | 0.386 |
| Heart Rate | 81.23±13.04 | 82.41±13.43 | 879731 | 0.026* |
| Liking Degree | 3.33±1.45 | 3.47±1.60 | 0.057 | 0.812 |
| Immersion | 2.93±1.28 | 4.40±1.18 | 10.621 | 0.003** |

\* p<0.05 , ** p<0.01

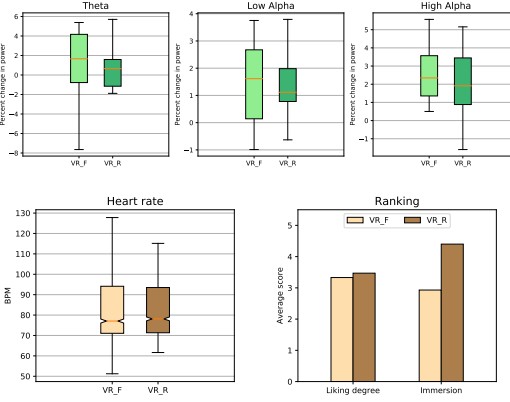

Figure 7: Physiological indicators and subjective scores in different virtual environments

Static and Dynamic Virtual Objects   The experimental results are shown in Fig. 8 and Table 3. In terms of brainwave signals, the two groups showed significant differences in the Theta band (p=0.038). Specifically, the mean value of Theta band elevation in the dynamic virtual object group (2.18) is significantly higher than that in the static virtual object group (0.20). In the Low and High Alpha bands, the two groups behaved differently but not significantly. In the pulse signals, the overall distribution of the two was very similar, and the mean heart rate of the dynamic group was slightly lower (p=0.006). In the subjective evaluation, the dynamic group scored slightly higher on liking and immersion than the static virtual object group.

Table 3: Variance analysis in different virtual objects

|  | Static | Dynamic | F | p |
|---|---|---|---|---|
| Theta | 0.20±3.08 | 2.18±4.07 | 4.516 | 0.038* |
| Low Alpha | 2.82±2.86 | 2.21±2.42 | 0.796 | 0.376 |
| High Alpha | 3.01±3.66 | 2.83±2.60 | 0.045 | 0.832 |
| Heart Rate | 82.53±12.14 | 81.57±11.20 | 599153 | 0.006** |
| Liking Degree | 3.67±1.54 | 4.47±1.36 | 2.275 | 0.143 |
| Immersion | 3.60±1.68 | 4.60±0.99 | 3.947 | 0.057 |

* p<0.05 , ** p<0.01

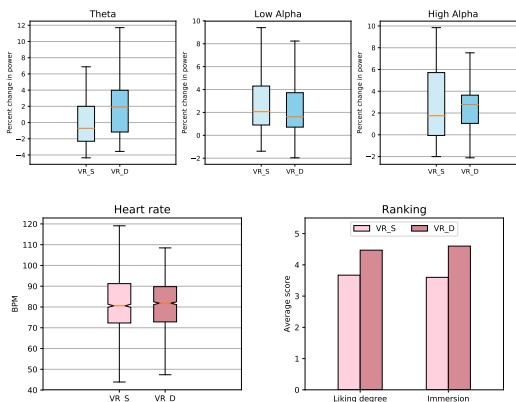

Figure 8: Physiological indicators and subjective scores in different virtual objects

In conclusion, from the perspective of physiological signals and subjective experience, we can think that the effect of mindfulness meditation on dynamic virtual objects is slightly better than that on static virtual objects. The increase of Theta band and decrease of heart rate in dynamic group represent better meditation, which proves that it is necessary to design a dynamic virtual object in the scene. Of course, the specific motion is determined according to the needs of the scene and task, which requires more in-depth research. In addition, some participants also reported having an easier visual center of gravity in the context of dynamic virtual objects, which may be helpful for concentration during meditation.

**Static and Dynamic Interactive Tasks** The experimental results are shown in Fig. 9 and Table 4.In terms of EEG signals, the improvement of static interaction task group in each frequency band was higher than that of dynamic interaction task group, and the Theta (p=0.029), Low Alpha (p=0.008) and High Alpha (p=0.005) bands showed significant differences. However, the dynamic interaction group had a lower overall heart rate distribution (p=0.005) and a wider range of heart rates in the upper and lower quartiles, indicating a greater reduction in heart rate before and after mindfulness meditation. From the subjective point of view, the static interaction group had higher scores of liking and immersion than the dynamic interaction group.

In summary, from the perspective of physiological signals, we can conclude that static interaction and dynamic interaction have different effects on mindfulness meditation. Different frequency bands of EEG in the static interaction group increase more, and the heart rate decreases faster in the dynamic interaction group. In terms of subjective experience, static interaction is slightly better than dynamic interaction. However, we also observed from interviews that this study has a strong correlation with individual experience.For the participants with meditation experience, they stated "I tap easier to concentrate" and "self-tapping woodfish can control the breathing

interval more precisely through visualization".But for those who didn't meditate, it might mean "I was a little fussy when I hit the fish at first" and "I would pay attention to whether I was in tune with the rhythm of the scene, and I would care if the rhythm was disturbed." Therefore, we should consider the audience and application scenarios when designing the interaction form.If in the design of a VR meditation application with different scenarios and tasks, two interaction modes can be designed for users to choose, and the dynamic interaction can also be considered as the pre-guidance of the static interaction.

Table 4: Variance analysis of EEG signals in different interactive tasks

|  | Static | Dynamic | F | p |
|---|---|---|---|---|
| Theta | 2.18±4.07 | -0.29±4.48 | 5.018 | 0.029* |
| Low Alpha | 2.21±2.42 | 0.48±2.43 | 7.652 | 0.008** |
| High Alpha | 2.83±2.60 | 0.84±2.69 | 8.527 | 0.005** |
| Heart Rate | 81.57±11.20 | 82.31±11.94 | 598287 | 0.005** |
| Liking Degree | 4.47±1.36 | 3.93±1.75 | 0.870 | 0.359 |
| Immersion | 4.60±0.99 | 4.13±1.68 | 0.857 | 0.362 |

* p<0.05 , ** p<0.01

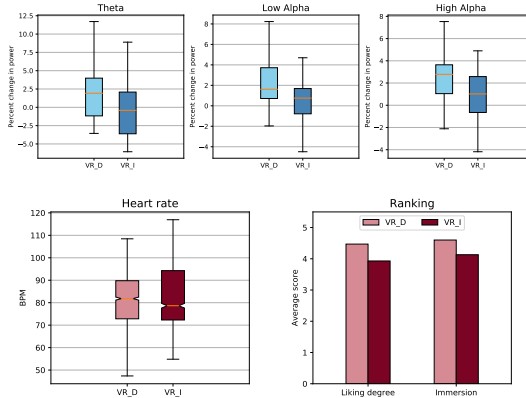

Figure 9: Physiological indicators and subjective scores in different interactive tasks

**System Evaluation** The results of BRUMS pre-test questionnaire and post-test questionnaire of all subjects were counted, and the mean and variance were calculated, as shown in Fig. 10 and Table 5.There was no significant difference in vigor between the samples before and after the experiment, while the mean value of tension decreased by 4.33(p=0.000) and the mean value of fatigue decreased by 3.67(p=0.009) after the experiment.Mean depression decreased by 2.67(p=0.009), mean confusion decreased by 4.47(F=13.153, p=0.001), and mean anger decreased significantly by 2.93(p=0.004).

In conclusion, after using this mindfulness meditation system, the emotions of tension, fatigue, depression, confusion and anger decreased significantly, indicating that the system established in this study can better improve negative emotions. Meanwhile, the vitality decreased slightly but not significantly, which was mainly caused by wearing VR equipment for a long time according to the interview.

## 6 CONCLUSION

This study aims to explore the influence of different designs on mindfulness meditation in virtual reality, hoping to provide guidance for future research on mindfulness application in VR. The main contents and achievements of this research are summarized as follows:

Table 5: Variance analysis of BRUMS

|  | Pre-test | Post-test | F | p |
|---|---|---|---|---|
| Tension | 1.80±2.14 | 6.13±2.88 | 21.891 | 0.000** |
| Vigor | 7.40±3.22 | 9.67±2.87 | 4.135 | 0.052 |
| Fatigue | 3.33±3.64 | 7.00±3.57 | 7.771 | 0.009** |
| Depression | 1.40±1.92 | 4.07±3.17 | 7.756 | 0.009** |
| Confusion | 2.60±2.35 | 7.07±4.15 | 13.153 | 0.001** |
| Anger | 1.20±1.70 | 4.13±3.18 | 9.921 | 0.004** |

* $p<0.05$ , ** $p<0.01$

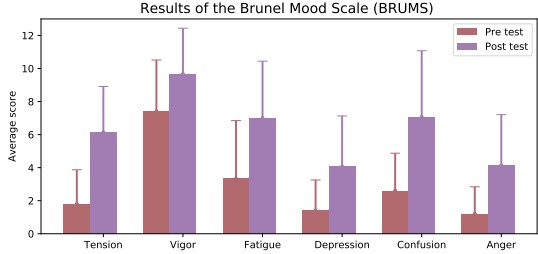

Figure 10: Pre-test and post-test BRUMS scores of the whole experiment.

First of all, we put forward the specific research questions and carried out the project design and development. The dynamic and static of mindfulness based on virtual reality is taken as the core research content of this study to explore the influence of different designs. From video and VR, flat and realistic virtual environment, static and dynamic virtual object, static and dynamic interactive task 4 research subproblems as the starting point for research and analysis. Through the design and development of different scenes, mindfulness meditation and interaction in VR are realized, while physiological measurement is carried out through biosensors.

We then conduct user experiments and produce applicable design guidelines. Fifteen participants were recruited to perform mindfulness meditation in six scenarios, and then physiological data collected in the experimental test and subjective questionnaire data were divided into four groups for processing and analysis. Finally, the design guide of mindfulness interactive system based on virtual reality was obtained:

(1) Mindfulness meditation in VR is objectively and subjectively better than mindfulness meditation in video, and meditation in VR can significantly make people more relaxed. If the venue and equipment permit, virtual reality equipment can be given priority to meditation.

(2) There is little difference in the overall mindfulness meditation effect between the flat style virtual environment and the realistic style virtual environment. The flat style virtual environment can be considered for devices with low hardware configuration, and for devices with better configuration and developers hoping to attract a wider range of users with immersion, the realistic virtual environment can be considered.

(3) The mindful meditation effect of dynamic virtual objects is slightly better than that of static virtual objects, and dynamic virtual objects can make people better enter the state of mindful meditation, so it is suggested to add dynamic virtual objects in the scene design.

(4) Static interaction and dynamic interaction have different influences on the effect of mindfulness meditation. The power of EEG signals increases more in static interaction, while the heart rate of participants decreases more rapidly in dynamic interaction. It is suggested that the meditation experience and application scenarios of users should be properly considered when designing interaction

forms, and different interaction tasks should be used according to different needs.

In conclusion, this study conducted a preliminary study on how to better design a mindfulness meditation system in VR, but there are still some shortcomings and future improvements in this study: First of all, the design in this study is just an example. There is a lot of imagination space to realize the design in VR, and there is still a long way to go to explore this direction. Secondly, in terms of hardware equipment, the VR headset used in the experiment has great room for improvement in weight and resolution, and more advanced VR and sensor equipment can be used for experiments in the future. At the same time, due to the limitation of equipment and space, participants are mainly recruited from schools, which can enrich the source of subjects in the future.

In general, studying how to better design the interactive system of mindfulness meditation in virtual reality can better bring mindfulness in virtual reality into the public eye and become a practical application product. It is hoped that future researchers can apply the virtual reality mindfulness meditation system in consumer products as soon as possible, so as to better improve people's emotions and psychological problems.

## ACKNOWLEDGMENTS

The authors wish to thank Tsinghua University Cross Innovation Workshop for providing equipment and space support.

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
