# OpenReview forum: "Research on the Influence of Different Designs on Mindfulness Meditation in Virtual Reality"
_graphicsinterface.org/Graphics_Interface/2023/Conference_SD — Submitted to GI 2023 - second deadline_

### Official Review · Reviewer_kStj · 2023-04-23
**Good idea but needs further revision**

**Rating:** 4
**Confidence:** 3

**Review:**

First of all, I would like to point out that the GI 2023 submission is double-blinded (https://graphicsinterface.org/conference/2023/cfp/). Thus, the authors should not reveal their authors and/or affiliations. Before making a submission, the authors should carefully read the venue's submission rules (as different ones have different rules). Though the anonymity rule is violated, I still provide my comments below.

- I appreciate that the authors carefully reviewed and presented relevant work in Sec 2. However, the authors did not try to relate the previous work to this work, such as what limitations exist in the previous work, how previous work inspired your work, did you chose your designs/study factors based on these works.

- Some namings are quite arbitrary. For example, the use of "flat style" and "realistic style". For my reading of sec 3.1, it seems that the distinction is solely based on their levels of detail (LOD, which is a specific term). Meanwhile, the levels of realism have different meanings and categorizations in different fields. I would suggest the authors do more research and avoid creating new ones unless necessary.

- The experiment part lacks important details and thus should be carefully revised. The authors did not distinguish between design and procedure and did not explain clearly the conditions and tasks. In addition to reading other study-based HCI work, I would recommend the authors to read section 8 of the book "human-computer interaction an empirical research perspective" first to get an idea of how to structure this part.

- The authors carefully reported their results, but the related discussion is limited. It would be nice if the authors could extend their discussions to talk about the implications behind the results, and/or what take-away messages that people can learn from the results.

In summary, I think the authors still need major revisions regarding the choice of design factors, the study description, and the discussion of the results. I would thus argue that it is not ready to be accepted. However, I would still want to point out that I do see the value of the work, and would definitely encourage the authors to re-submit elsewhere after revisions.

---

### Official Review · Reviewer_fGVR · 2023-04-24
**good topic, but main concerns in study design/writing**

**Rating:** 4
**Confidence:** 3

**Review:**

This work presents an experiment on mindfulness meditation in VR, focusing on three aspects: 1) virtual environments, 2) virtual objects, and 3) interactive tasks. The general flow is well-structured, the data analysis is carefully presented, and I found some interesting discussions in the analysis and explanation of study results. I can tell from the supplemental video that the implementation quality is quite high.


However, I have a few major concerns that make me feel this manuscript is not ready to publish at its current stage:


Major concerns:
• The Related Work section covers a large amount of work but barely explains how this presented work can be positioned in the literature. The authors are suggested to include a summary to clearly identify limitations of prior work and how this work fills the gap in the literature.
• This issue is related to my second concern, that the study design is a bit broad, and it's not clear whether (and how) these focused aspects are missing in the literature or need further investigation. Personally speaking, each individual aspect is challenging enough to investigate since each of them involves a lot of dimensions (e.g., moving patterns of objects and number of objects in aspect 2 "Virtual Object Design," or which body part is tracked and pace of movement in aspect 3 "Interactive Task Design"). A broad exploratory study described in this work is informative but could provide limited research contribution to the community.
• The study condition: "virtual environment design" (flat natural pathways in the countryside vs. realistic natural trails by the river) involves extra parameters and weakens the proposed claim.
• The study procedure: based on Fig. 4, it is unclear whether 1) the participants are divided into 2 groups (then it's unclear how this division took place) or 2) the participants took the roles of all groups and scenarios (then they might experience unknown accumulated mediation effects across trials).
• Section 5.1 is a bit wordy and could be condensed to highlight key decisions as these aspects are not the key contributions of this work. Plus, there are placeholder texts left unfixed in the second paragraph.
• Section 5.2: please also report the degree of freedom when reporting F-values.

About anonymity: The manuscript, including the authors' field, contact info, content (e.g., participants' background and experiment location, etc. in Sec. 4), and acknowledgement, as well as the supplementing video, shall be carefully anonymized based on the submission instructions. This information can only be released in a camera-ready submission. Please be aware that this work might be desk rejected as multiple parts of this submission violate the anonymity policy.

To sum up, I am arguing for a weak rejection. The authors are encouraged to further improve the design & presentation of this work.

Minor concerns:
• Some quotes/claims in the Introduction are missing citations.
• Section 2.1 is a bit wordy and could be condensed.
• The first paragraph in Section 2.3 starts from a few claims (e.g., "they have analyzed …") without proper citations.
• There are some minor grammar issues in Sec. 3.1.
• I found some misplacing punctuations in the manuscript. Please proof-read.

---

### Official Review · Reviewer_Mzyo · 2023-05-01
**Interesting research direction, but design and execution of study should be reconsidered and better grounded in existing literature**

**Rating:** 4
**Confidence:** 3

**Review:**

The authors of this paper investigate how different designs influence mindfulness meditation in a virtual reality environment. To achieve this, they report on a study design and execution in which they collected data about participants' heart rate, EEG sighnals, and subjective data.

The paper is relatively easy to follow, despite a few typos and language errors (e.g., "We then conduct user experiments and produce applicable design guidelines." Although the introduction lacks references to the literature in its first part, the related work is interesting and I definitely learned new things by reading it. That being said, the related work is not used to justify the presented approach, therefore making it difficult for the reader to identify/assess how the presented work differs from the literature.

While the subquestions do make sense and the study design reflects these questions, it creates a major problem with the work: its scope is very large. With so many design aspects being investigated simultaneously, it is difficult to draw any conclusion from the study given at the same time the large number of conditions, the study design, and the relatively small number of participants. It also means that many design decisions were made that could have been made differently, and this results in a large number of factors to analyze simultaneously. Given the complexity of the design, I am left thinking that a qualitative approach to studying the different design parameters would be more appropriate.

The last part of the paper feels unfinished. It jumps directly to the conclusion from the results, where I would expect a discussion that puts the results in perspective, and relates to the literature. Again, it makes it difficult to know how the results are different/complementary to existing work. Such a discussion should also highlight higher-level themes from the results, limitations of the approach/study, and guidance for research in this area.